# Onset of double subduction controls plate motion reorganisation

Kuidi Zhang [1], Jie Liao [1,2,3] ✉ & Taras Gerya [4]

Face-to-face double subduction systems, in which two oceanic plates subduct toward each other, are essential elements of plate tectonics. Two subduction zones in such systems are typically uneven in age and their spatially and temporally variable dynamics remain enigmatic. Here, with 2D numerical modelling, we demonstrate that the onset of the younger subduction zone strongly changes the dynamics of the older subduction zone. The waxing younger subduction may gradually absorb plate convergence from the older one, resulting in older subduction waning featured by the dramatic decrease in subduction rate and trench retreat. The dynamical transformation of subduction predominance alters the intraplate stress and mantle flow, regulating the relative motion among the three different plates. The process of waxing and waning of subduction zones controls plate motion reorganisation, providing a reference to interpret the past, present, and future evolution of several key double subduction regions found on the modern Earth.

Subduction is an essential process in the evolution of Earth[1], driving plate tectonics[2], shaping Earth surface[3], and triggering mantle convection[4]. The dynamic evolution of subduction zones varies temporally and spatially, featured by the initiation of new subduction zones and the termination of the older ones[3,5]. An interesting phenomenon is that the younger subduction zones pervasively initiated adjacent to the pre-existing older ones[3,5,6]. One specific setting with such phenomenon is the face-to-face double subduction zones[7]. Three examples of face-to-face double subduction presented in nature[5] are the Manila and Philippine subduction zones[8–10] (Fig. 1a), the Vitiaz-Tonga and Vanuatu subduction zones[11,12] (Fig. 1b), and the Central America-Colombia and Lesser Antilles subduction zones[13–15] (Fig. 1c), all of which have experienced similar tectonic evolution (see below)[8–15] and present similar observed features (Fig. 1)[16–19], implying a common pattern that the development of the younger subduction zones inhibits the older ones. However, the dynamic waxing and waning of subduction zones in the face-to-face double subduction systems remain enigmatic.

In the face-to-face double subduction systems, the younger subduction zone initiated adjacent to the pre-existing older one with time intervals of several to several tens of Myrs[5,6]. One overriding plate shared by the two subductions is present in each face-to-face double subduction system, and the initiation of the older subduction was driven by the overthrusting of the overriding plate in the early stage, while the younger subduction developed through the active motion of the subducting plate[8–15]. The younger subduction dominated the evolution and gradually inhibited the older one, as shown by the observations[16–19] (Fig. 1): (1) The initiation of the Philippine subduction (8–9 Ma)[20] is younger than the adjacent Manila subduction (-18 Ma)[21] located to the west. Since the onset of the younger Philippine subduction, the activity of the southern segment of the Manila subduction, which overlaps the Philippine subduction, has decreased dramatically, as evidenced by the slow trench retreat[17,18] and low convergence rate[22] (Fig. 1a). (2) The northeastward trench retreat of the northern segment of the Vitiaz-Tonga subduction experienced episodic interruption by the adjacent younger subductions (including the youngest Vanuatu subduction), which eventually led to the extinction of the Vitiaz subduction[11] (Fig. 1b). (3) The Central America-Colombia and Lesser Antilles double subduction zones present similar features, marked by the

[1]School of Earth Sciences and Engineering, Sun Yat-Sen University, Guangzhou, China. [2]Southern Marine Science and Engineering Guangdong Laboratory (Zhuhai), Zhuhai, China. [3]Guangdong Provincial Key Lab of Geodynamics and Geohazards, Guangzhou, China. [4]Department of Earth Sciences, Swiss Federal Institute of Technology Zurich, Zurich, Switzerland. ✉e-mail: liaojie5@mail.sysu.edu.cn

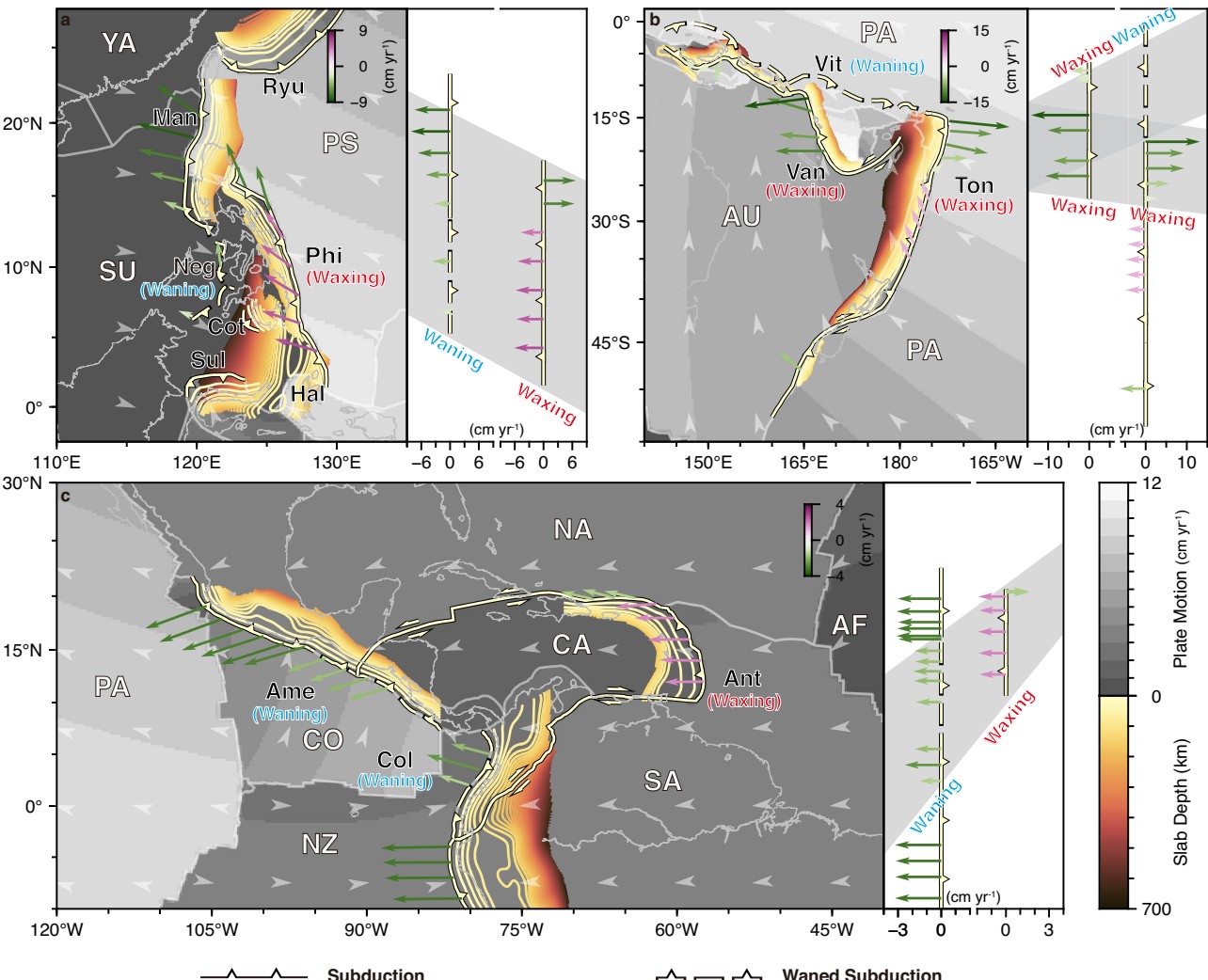

**Fig. 1 | Three natural examples showing waxing and waning in face-to-face double subduction zones and plate motion reorganisation.** Initiation of the younger Philippine subduction (**a**), the Vanuatu subduction (**b**), and the Lesser Antilles subduction (**c**) inhibits the older Manila subduction, the Vitiaz subduction, and the Central America-Colombia subduction, respectively. The base maps include the rate (grey colours) and direction (grey arrowheads) of plate motion[16,18]. The thick lines with triangles indicate the trenches. The vector arrows indicate the trench migration, with the colour indicating the magnitude of the velocity, and the positive/negative values representing trench advance/retreat[17,18]. The contour lines constrain slab morphology[19]. The right columns show the magnitude of trench migration and the grey areas represent the regions of double subduction. Plates:[65] AF-African; AU-Australian; CA-Caribbean; CO-Cocos; NA-North-American; NZ-Nazca; PA-Pacific; PS-Philippine Sea; SA-South American; SU-Sunda; YA-Yangtze. Subduction zones:[4,11,12,19] Ame-Central America; Ant-Lesser Antilles, Col-Colombia; Cot-Cotabato; Hal-Halmahera; Man-Manila; Neg-Negros; Phi-Philippine; Ryu-Ryukyu; Sul-Sulawesi; Ton-Tonga; Van-Vanuatu; Vit-Vitiaz.

decrease in trench migration[17,18] and steeper slab of the Central America-Colombia subduction[19] and slower overriding plate motion (Caribbean Plate) in the overlapping region[13,23] (Fig. 1c). These three natural examples imply a common pattern that the onset and waxing of the younger subduction zones lead to the waning of the older ones, and the waxing and waning of subductions control plate motion reorganisation.

We propose that the onset and waxing of the younger subduction and the waning of the older subduction in the face-to-face double subduction is a common phenomenon controlling plate motion reorganisation. In this study, we explore numerically how the onset of double subduction controls plate motion reorganisation. We address that, based on the modelling results, the onset and development of the younger subduction inhibit the older one through plate motion partitioning. Our model results capture the first-order geodynamic characteristics of plate motion reorganisation through the waxing and waning of the double subduction and shed light on the evolution of natural examples.

## Results

### Onset and evolution of face-to-face double subduction

We set up 2D thermomechanical numerical models (Methods and Supplementary Fig. 1) to simulate the dynamic evolution of face-to-face double subduction (Fig. 2). The two subductions initiated successively at the initially imposed weak zones. Figure 2 shows the critical moments of reference model evolution, which can be subdivided into three characteristic stages.

The older subduction develops in the early stages forming the typical single subduction mode (Fig. 2a). The overthrusting of the overriding plate predominantly drives the subduction, featured by the compressional stress state in the overriding plate (Fig. 2e). The trench retreats passively with the same speed as the overriding plate (7 cm yr$^{-1}$) and the dipping angle of the subduction slab is moderate. The sinking of the subduction slab triggers counterclockwise poloidal flow in the asthenospheric mantle (Fig. 2e).

Face-to-face double subduction is established, marked by the initiation of the younger subduction after 5 Myrs (Fig. 2b, f).

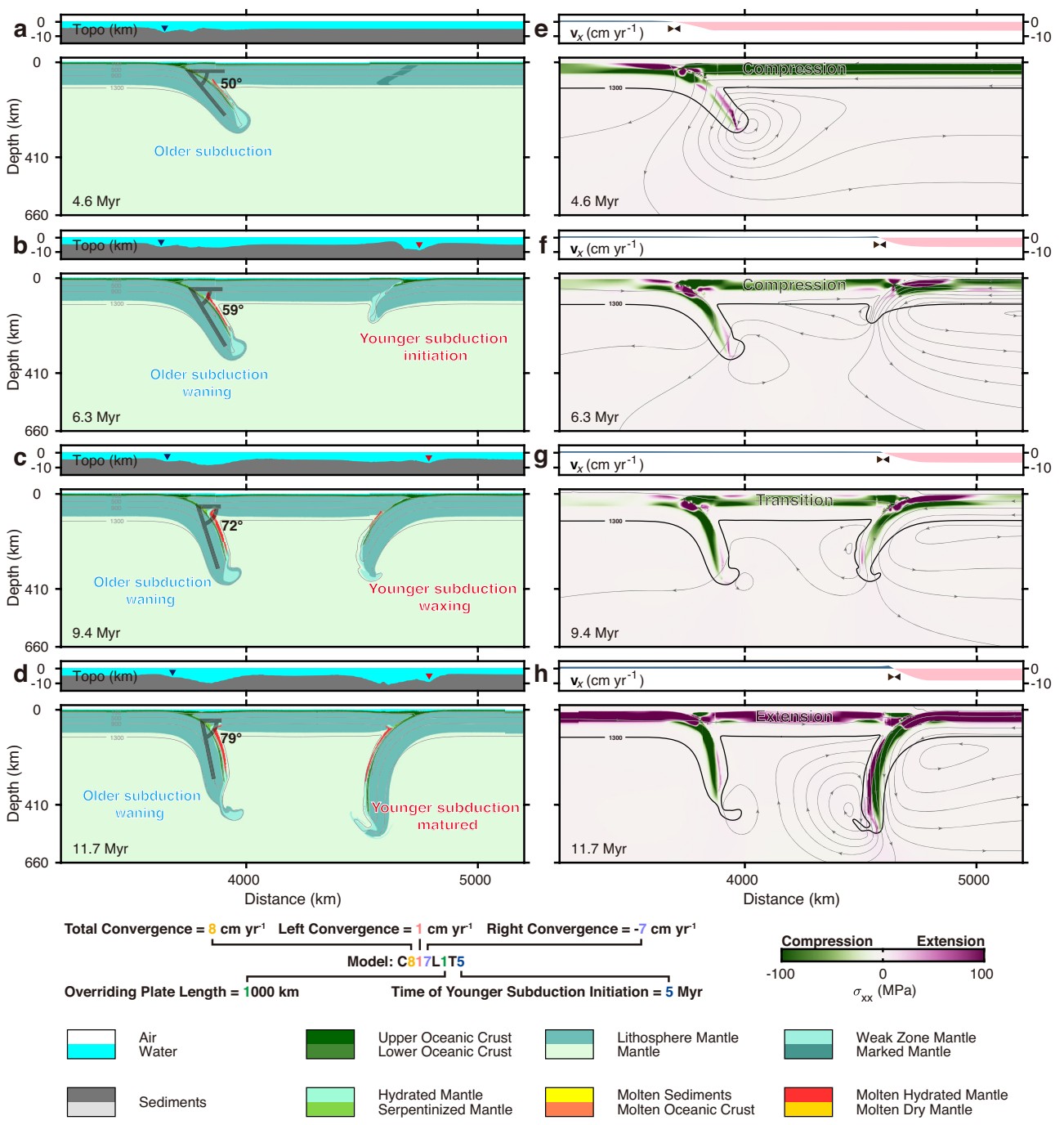

**Fig. 2 | Reference model C817L1T5 (labelling logic see the legend) results showing the dynamic evolution of waxing and waning in face-to-face double subduction zones. a–d** Top: topography snapshots with dark blue and red triangles marking the trenches of the older and younger subductions, respectively; bottom: lithology snapshots with temperature contours (grey lines). **e–h** Top: averaged horizontal velocity of the plates reflecting the main regions (double black triangles) absorbing convergence; bottom: stress evolution with temperature contours outlining the slab (1300 °C, thick black lines) and the contours of stream function (thin grey lines with arrowheads) showing plate motion and mantle convection.

The evolution of the younger subduction is predominantly controlled by subduction plate advance, with negligible trench migration (Fig. 2b). Initiation of the younger subduction triggers intense horizontal mantle flow towards the older subduction slab (Fig. 2f). As the younger subduction slab steepens, the horizontal mantle flow gradually transforms into vertical mantle flow between the two slabs (Fig. 2c, g). The stress fields record the transition from compression to extension of the overriding plate due to the increased slab pull of the younger subduction (Fig. 2f, g). The younger subduction strongly absorbs convergence and prevents stress transmission

from its lower plate to the overriding plate, resulting in a decrease in trench migration of the older subduction (Fig. 2b, c, f, g). Thus, the younger subduction dominates the model evolution.

The younger subduction further facilitates vigorous vertical mantle flow between the two slabs (Fig. 2d, h). Meanwhile, the overriding plate is completely under extensional stress (Fig. 2h), indicating that the younger subduction has become self-sustaining. Thus, once the younger subduction initiated, the activity of the older subduction decreases dramatically (e.g., termination of trench retreat), resulting in the waxing and waning of the younger and older

subductions and plate motion reorganisation in the face-to-face double subduction system.

The effect of essential physical parameters (i.e., the partitioning of plate convergence between the two subduction zones, the lengths of the overriding plate, and the onset time of the younger subduction) on the model evolution has been systematically investigated (see Supplementary Figs. 2–13 for plate motion and lithospheric stress evolution). The model results are categorised into three types with respect to the evolution of younger subduction (Fig. 3), i.e., double subduction dominated by younger subduction (Figs. 2, 3f), transitional double subduction with abandoned younger subduction (Fig. 3g, Supplementary Fig. 2), and single subduction (Fig. 3h, Supplementary Fig. 3). The force analysis reveals the competition in subduction predominance between the older and younger subductions (Fig. 3i–k). At the onset of older subduction, plates in all models are in the compressional state. After 5 Myr, the onset of double subduction begins. In the ensuing competition between older and younger subduction, $F_{right}$ is higher than $F_{left}$, implying that the double subduction system is dominated by the younger one (Fig. 3i). Otherwise, $F_{left}$ is higher than $F_{right}$, indicating that the younger subduction has failed to compete with the older subduction. Finally, the system is dominated by the older subduction (Fig. 3j). The $F_{left}$ is higher after 5 Myr without the competition stage, indicating single subduction pattern (Fig. 3k).

The partitioning of the plate convergence rate between the two subduction zones is the predominant factor influencing model evolution, i.e., a large plate convergence rate promotes fast subduction

(Fig. 3a, Supplementary Figs. 4, 5). The younger subduction with a large plate convergence rate evolves rapidly and dominates evolution in the double subduction model, resulting in subduction waxing and waning (Fig. 3f). With the decrease in the overriding plate convergence rate, the younger subduction initiates but fails to develop, and plate convergence is largely absorbed by the older subduction (Fig. 3g, Supplementary Fig. 2). The younger subduction fails to initiate with further decrease in the overriding plate convergence (Fig. 3h, Supplementary Fig. 3). Similar plate reorganisation can also be perceived in the double subduction systems that the subduction polarities of the two slabs are identical[24,25]. Varying the length of the overriding plate (Fig. 3b, c, Supplementary Figs. 6–9), and the total of convergence rate (Fig. 3d, Supplementary Figs. 10, 11) do not change the first-order features of the models evolution. The onset of the younger subduction (Fig. 3e, Supplementary Figs. 12, 13), however, affects model evolution since it determines the subduction duration of the older one, i.e., the older subduction becomes self-sustaining with longer subduction duration, inhibiting the evolution of the younger one.

## Plate motion reorganisation through double subduction

Dynamic evolution of the overriding plate reveals the waxing and waning of subductions in the face-to-face double subduction systems (Fig. 4). Horizontal motion of the overriding plate initiates and drives the older subduction associated with the trench migration in the early stage, prior to the onset of the younger subduction (Fig. 4a). The overriding plate is slightly slower than plate convergence, indicating a

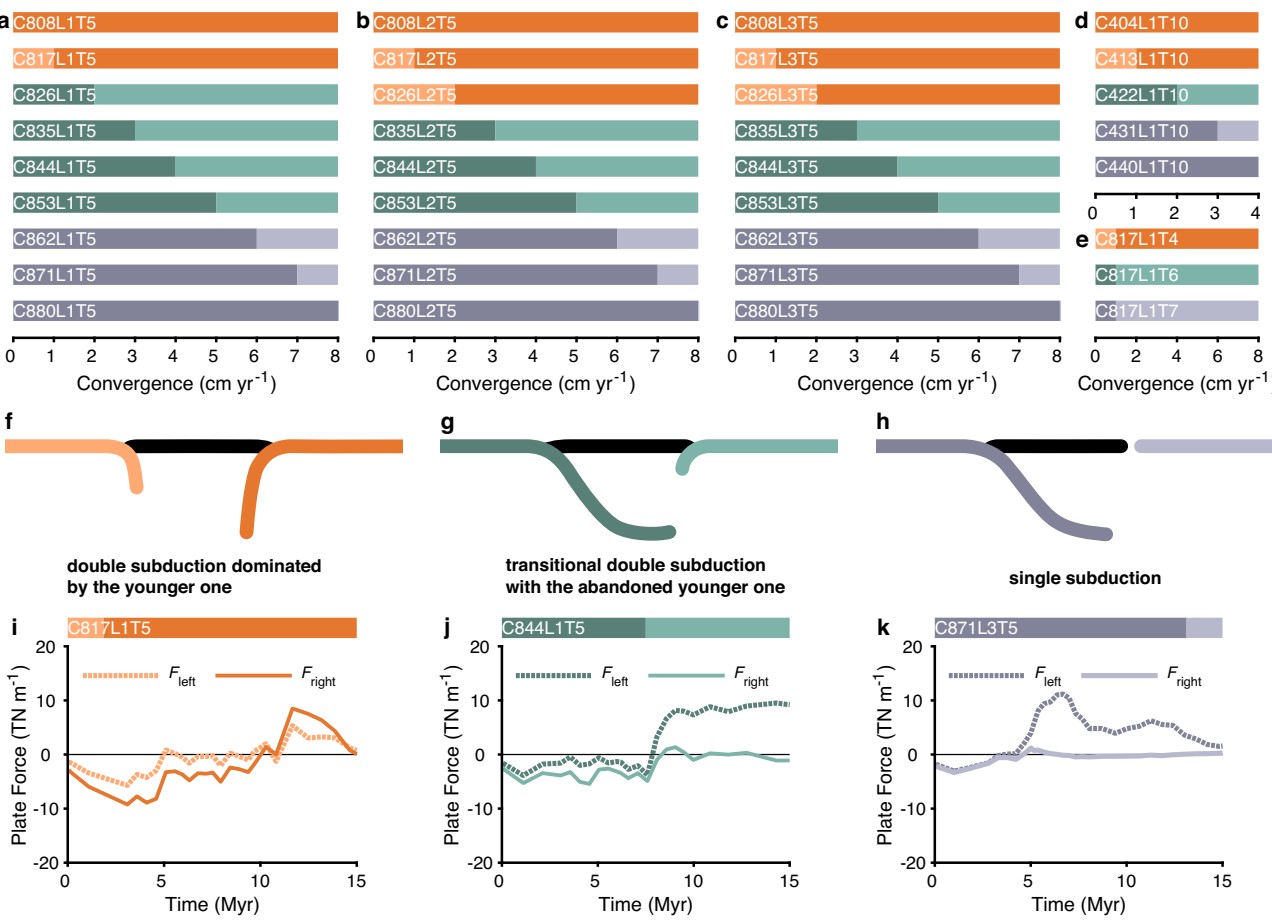

**Fig. 3 | Regimes of face-to-face double subduction. a–c** Model results (labelling logic see Fig. 2) with different lengths of the overriding plate, i.e., 1000 km, 2000 km, and 3000 km (total plate convergence rate of 8 cm yr⁻¹ and onset of the younger subduction at 5 Myr). **d** Model results with lower total plate convergence

rate (4 cm yr⁻¹) and later onset of younger subduction (10 Myr). **e** Model results with varied onset time of the younger subduction. **f–h** Three patterns of model evolution. **i–k** Force analysis of the three patterns of model evolution ($F_{left/right} = \int_{z_{LAB}}^{z_{surf}} \sigma_{xx} dz$, negative during compression and positive during extension).

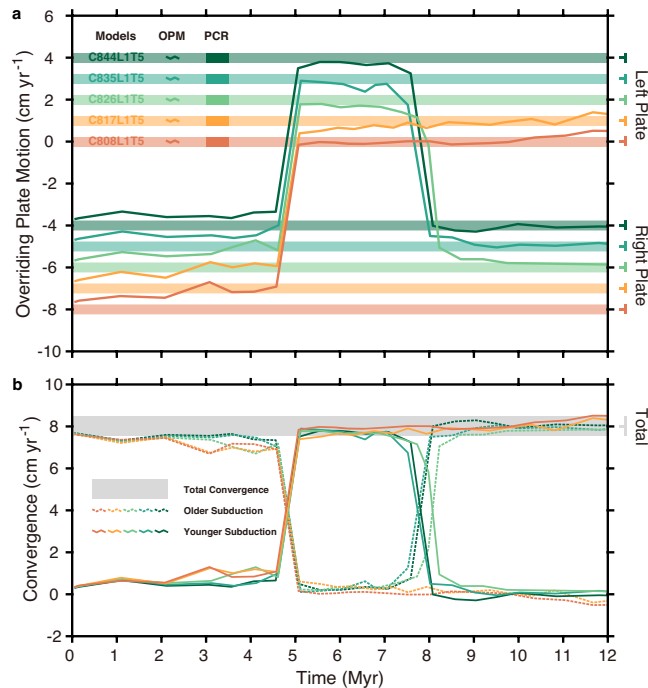

**Fig. 4 | Reorganisation of plate motion in face-to-face double subduction systems. a** Dynamic motion of the overriding plate. The solid lines indicate the overriding plate motion (OPM) and the horizontal bands mean the plate convergence rates (PCR) of the left and right plates. **b** Partitioning of plate convergence is absorbed alternatively by the older (dashed lines) and younger subductions (solid lines). The horizontal grey band means the total convergence.

small amount of intraplate compression (Fig. 4a). After the onset of the younger subduction (5 Myr), the overriding plate moves in the opposite direction (Fig. 4a), because the younger subduction absorbs nearly the total plate convergence rate (Fig. 4b). The older subduction becomes abandoned with the rapid declining in trench retreat due to the reducing of convergence. The direction of the overriding plate motion reversed one more time in the transitional double subduction models (C826L1T5, C835L1T5, and C844L1T5), due to the abandonment of the younger subduction and the reactivation of the older one (Fig. 4 Supplementary Fig. 2). This evolution pattern is affected by the partitioning of plate convergence. There is no obvious deformation in the overriding plate (e.g., back-arc spreading), so the rate of trench retreat could be equal to the motion of the overriding plate[17]. Trench retreat of the older subduction decreases dramatically after the onset of younger subduction because overthrusting from the overriding plate vanishes.

## Discussion

### Plate motion reorganisation in natural double subduction

The Manila and Philippine face-to-face subductions system is one key example. The evolution of the Manila subduction with trench retreat is driven by NW overthrusting of the Philippine Sea plate since -18 Ma[26]. The Philippine subduction initiated to the east of the Manila subduction with opposite subduction polarity at -8 Ma[20], triggered by the collision between the Palawan continent and the Luzon island arc[9] (Fig. 5a). The Philippine subduction developed and propagated laterally[9,20], absorbing plate convergence and leading to the dramatic decay of the Negros subduction in the Sulu Sea[8] (Fig. 5b). Trench retreat of the south Manila-Negros-Cotabato subduction is extremely slow at present[17,18], with large slab dip angles[10,19] (Figs. 1a, 5c). Thus, the waxing of the Philippine subduction resulted in the waning of the south Manila-Negros-Cotabato subduction, and the dynamic evolution

can be approximated by our model results (Fig. 2a–c, e–g). Furthermore, we propose that with the possibility of the northward propagation of the Philippine subduction (probably along the East Luzon trough[27,28], Fig. 5c), the entire Manila subduction zone will become extinct, as plate convergence is absorbed by the Philippine subduction and the young and warm South China Sea plate[21] may not provide sufficient slab pull (Fig. 5d). It is likely that the Philippine subduction will connect with the Ryukyu subduction, and a new back-arc basin will form in the overriding plate due to slab rollback in the future, by which time the South China Sea basin may not be closed with the extinction of the Manila subduction (Figs. 2d, h, 5d).

Multiple subduction events with waxing and waning between the older and younger subductions are recorded in the face-to-face Vanuatu and Vitiaz-Tonga subduction system, which is located between the Pacific plate and the Australian plate[11] (Fig. 5e). The most recent event is the replacement of the older Vitiaz subduction by the younger Vanuatu subduction (Fig. 5e, f). The Vitiaz subduction was governed by the northward overthrusting of the Australian plate, which was interrupted by the input of the Ontong Java Plateau into the subducting plate[11,12] (Fig. 5e), and the younger Vanuatu subduction initiated a few million years later by the absorption of the N-S convergence of the Australian plate[11,12] (Fig. 5f, g). The eastern and western segments of the younger Vanuatu subduction developed rapidly (while the central part was largely inhibited by the input of the Australian continental margin) featured by the fast trench retreat[11,16,18] (Figs. 1b, 5e–g). Since the offset distance between the Vitiaz-Tonga and Vanuatu subductions is small, intensive mantle flow may be regulating the regional subduction evolution[29]. The Vanuatu subduction may propagate southwards in the future[28] (Fig. 5h), continuously absorbing the convergence from the Australian plate. However, unlike the Philippine subduction, the younger Vanuatu subduction may not replace the older Tonga subduction because of the slower convergence rate of the Australian plate (i.e., highly oblique motion) and the faster convergence rate of the Pacific plate.

The Central America-Colombia and Lesser Antilles face-to-face subduction system records subductions waxing and waning[13]. In contrast to the North and South American plates, which overthrust towards the East Pacific plate forcing single subduction with rapid trench retreat[17,18] and flat slabs[19,30] (Fig. 1c), the Central America region developed face-to-face double subduction since the Late Cretaceous[13–15] (Fig. 5i–k). The convergence rate of the Central America-Colombia subduction decreased (i.e., 12.3 cm yr⁻¹ at 49 Ma, 7.4 cm yr⁻¹ at 22 Ma and 6.1 cm yr⁻¹ at present) after the initiation of the Lesser Antilles subduction, whose convergence rate increased (i.e., -1.5 cm yr⁻¹ at 49 Ma, -1.2 cm yr⁻¹ at 22 Ma and 2.2 cm yr⁻¹ at present)[3,16,18]. Waxing of the Lesser Antilles subduction may become more significant in the future with the possibility of northward propagation[28], which can inhibit the southern North American plate motion and the North-Central America subduction (Fig. 5l). The initiation of Lesser Antilles subduction at lower plate convergence rates suggests that additional tectonic processes are involved, such as plume facilitated subduction initiation proposed in previous studies[13–15].

Onset and evolution of the double subduction system often involved complex tectonic processes, such as collision[31–34] and plume[34–37], which make the waxing and waning of subduction even more enigmatic. Collision could anchor slab retreat forming a curved trench geometry according to previous 3D geodynamical models[38–40]. This scenario has occurred at the southern end of the Manila subduction[41], and collision may laterally inhibit Manilia subduction, but its influence may be less important due to the small length of the curved trench at the southern end. The Collision between the Ontong Java oceanic plateau and volcanic arcs has prevented Vitiaz subduction and induced Vanuatu subduction[31–33]. Furthermore, we suppose that the absorption of plate convergence by the younger

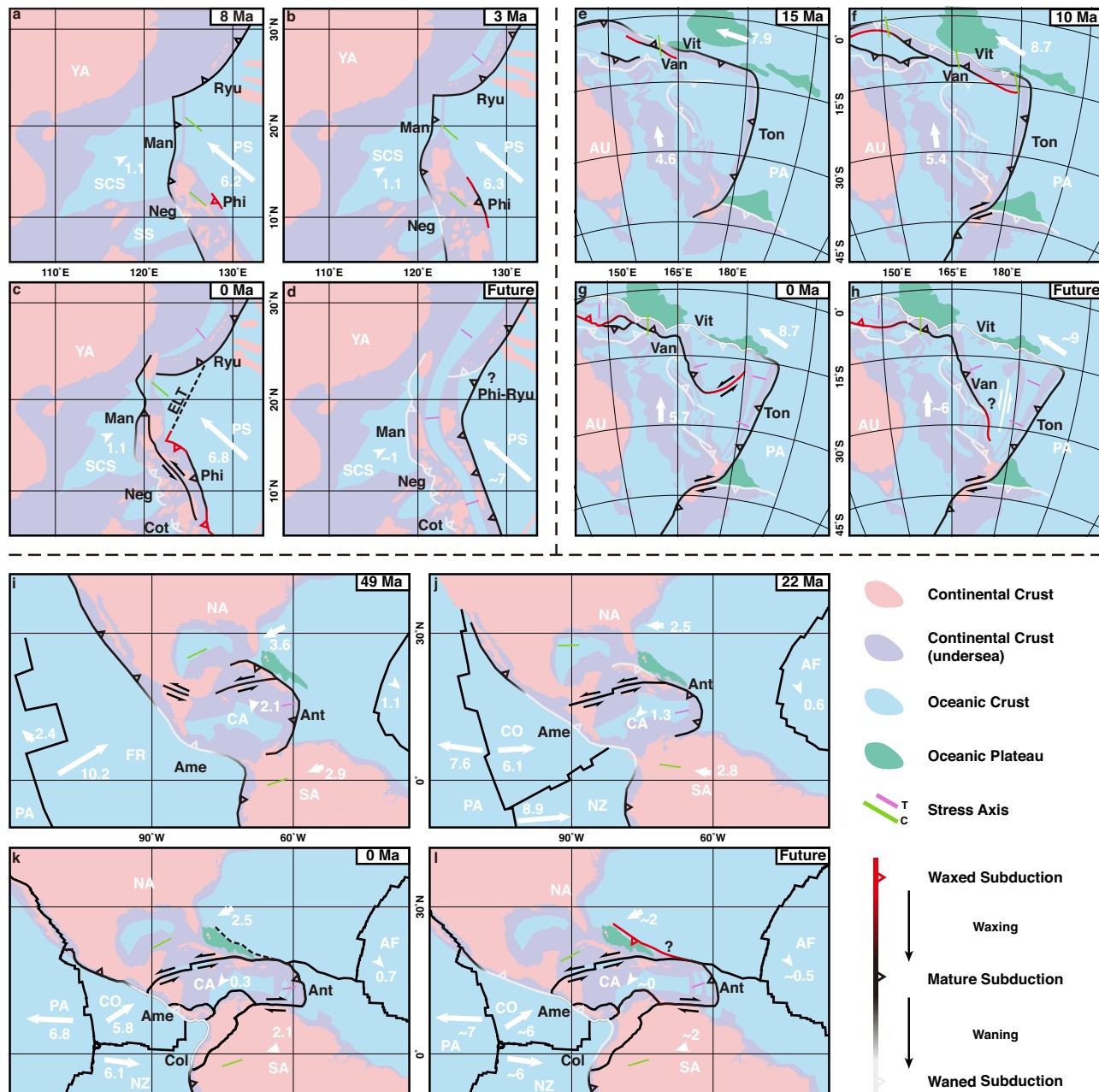

**Fig. 5 | Plate reorganisation in the past, present, and future due to subduction waxing and waning of the three natural examples. a–d** Manila and Philippine subductions[8]. **e–h** Vanuatu and Vitiaz-Tonga subductions[11]. **i–l** Central America-Colombia and Lesser Antilles subductions[13]. White arrows and numbers represent the direction and magnitude (cm yr⁻¹) of plate convergence, respectively[3,17,18]. The short bar indicates the direction of stress (C-compression, T-tension). Solid lines show the plate boundaries, with double arrows representing the transform fault and triangles representing subduction, with colours marking the waxing and waning of subduction. Dashed lines indicate the potential propagation direction of the subduction zone. SCS-South China Sea plate, SS-Sulu Sea plate, ELT-East Luzon trough, FR-Farallon plate. Other abbreviations are explained in Fig. 1.

Vanuatu subduction may promote the waning of Vitiaz subduction, which may decline with the failed input of oceanic plateau[42]. Although oceanic plateau collision and plume may induce the complex double subduction in Central America[34,36], how they influence subduction waxing and waning is difficult to assess quantitatively due to the poorly constrained spatio-temporal evolution of the collision and plume[14].

The initiation and termination of subduction zones has occurred repeatedly in the tectonic Earth, as revealed by plate reconstructions, providing us with a clear framework of Earth surface motions[3,43]. Complex subduction systems (e.g., double subduction zones) are established, which determine regional geodynamic evolution. The face-to-back double subduction models (i.e., identical subduction polarity) were previously established to explain the rapid convergence between the India and Eurasia plates[44,45], and the change in trench migration of the Izu-Bonin-Marianas subduction[46,47]. The back-to-back double subduction models (i.e., opposite subduction polarity) have been studied to demonstrate the closure of oceanic basins[48,49]. In addition, previous modelling studies of face-to-face double subduction models have explored vigorous mantle upwelling[34,50,51], anomalous pressure changes[7,52], and slab morphology[53]. Our model results provide a reference for interpreting the waxing and waning of subduction zones in face-to-face double subduction systems. Waning subduction is typically characterised by passive slab sinking and trench

retreat forced by the motion of the overriding plate, which accumulates intensive compressional stress. The waxing subduction is featured by active slab sinking and trench retreat leading to extension in the overriding plate. The waxing and waning of subductions control plate motion reorganisation and dominate regional geodynamics.

## Methods

### Code and governing equations

This study used the 2D petrological-thermomechanical code I2VIS[54] to simulate the face-to-face double subduction. The code employs the finite difference method to solve the mass, momentum, and energy conservation equations on a staggered non-uniform Eulerian grid and the non-diffusive marker-in-cell technique to transport physical properties by Lagrangian markers:

$$\text{div}(\mathbf{v}) = 0, \tag{1}$$

$$\text{div}(\boldsymbol{\sigma}') = \text{grad}(P) - \rho\mathbf{g}, \tag{2}$$

$$\rho C_p \frac{DT}{Dt} = \text{div}(k \cdot \text{grad}(T)) + H_r + H_a + H_s + H_L, \tag{3}$$

where $\mathbf{v}$ is the velocity vector, $\boldsymbol{\sigma}'$ is the deviatoric stress tensor, $P$ is the dynamic pressure, $\rho$ is the density, $\mathbf{g}$ is the gravitational acceleration, $C_p$ is the heat capacity, $T$ is the temperature, $k$ is the thermal conductivity, $H_r$, $H_a$, $H_s$ and $H_L$ are radiogenic, adiabatic, frictional, and latent heating source, respectively, and $\frac{DT}{Dt}$ represents the substantive time derivative of the temperature at the Lagrangian point.

Visco-plastic non-Newtonian rheologies are used in the model, accounting for eclogitisation in the oceanic crust and olivine-spinel (410 km) and spinel-perovskite (660 km) phase transitions. In addition, the model includes melting, hydration/dehydration, surface processes, and strain weakening[55,56].

### Model setup

The 10,000 × 1520 km model (Supplementary Fig. 1) is resolved by 1401 × 511 non-uniformly distributed Eulerian nodes, with the horizontal grid spacing varying from 10 km at the side boundaries to 4 km in the centre and the vertical grid spacing varying from 5 km at the bottom boundary to 1 km at the surface. Approximately 16 million randomly distributed Lagrangian markers are imposed. The right, left and top boundary conditions are free-slip and the bottom is permeable, which could reduce the boundary effect on mantle convection. The 20 km thick air layer and 5 km thick water layer are used to form free-surface (sticky air/water)[57]. The oceanic crust is 8 km thick and consists of 3 km of basalt and 5 km of gabbro. The thermal structure of the oceanic lithosphere is described by the plate cooling model[58] from 273 K at the surface to 1573 K at the lithosphere-asthenosphere boundary with a plate thickness of 95 km and a cooling age of 40 Myr. Temperature of the mantle beneath the lithosphere-asthenosphere boundary increases linearly with a gradient of 0.5 K km$^{-1}$.

The entire oceanic plate was divided into three sections by two weak zones with opposite polarities. Two weak zones are prescribed in the model and weakened successively to induce two subductions. The left weak zone is activated initially to induce the older subduction, and the right weak zone is activated after a certain time (e.g., 5 Myr in the reference model) to facilitate the formation of the younger subduction. The two velocity boundaries are imposed at a distance of 2000 km from the left and right boundaries to drive plate convergence. The effect of different plate convergence rates partitioning in double subduction is systematically tested. Two low-viscosity zones are imposed along the left and right sides of the oceanic plates to simulate spreading ridges as plates move away from the lateral boundaries.

### Rheological model

Viscous and plastic rheology are applied in the model. Ductile deformation takes into account dislocation and diffusion creep, which depends on strain rate, pressure ($P$), and temperature ($T$)[59]:

$$\eta_{\text{diff}} = \frac{1}{2} A \exp\left(\frac{E + VP}{RT}\right) \sigma_{\text{cr}}^{1-n}, \tag{4}$$

$$\eta_{\text{disl}} = \frac{1}{2} A^{\frac{1}{n}} \exp\left(\frac{E + VP}{nRT}\right) \dot{\varepsilon}_{\text{II}}^{\frac{1}{n}-1}, \tag{5}$$

$$\eta_{\text{ductile}} = \left(\eta_{\text{diff}}^{-1} + \eta_{\text{disl}}^{-1}\right)^{-1}, \tag{6}$$

where $\sigma_{\text{cr}} = 3 \times 10^4$ Pa is assumed diffusion-dislocation transition stress, $\dot{\varepsilon}_{\text{II}} = \sqrt{1/2\dot{\varepsilon}_{ij}^{2}}$ is square root of the second invariant of the strain rate tensor $\dot{\varepsilon}_{ij}$ and $A$ (pre-exponential factor), $E$ (activation energy), $V$ (activation volume) and $n$ (stress exponent) are the experimentally determined flow law parameters. $R$ is gas constant (Supplementary Table 1).

Plastic rheology is used to limit the upper limit of viscous (ductile) deformation as follows:

$$\eta_{\text{ductile}} \leq \frac{c + \mu P}{2\dot{\varepsilon}_{\text{II}}}, \tag{7}$$

where $c$ is the rock cohesion at $P = 0$, $\mu$ is the effective internal friction coefficient.

### Density Model

The density of rocks varies with pressure ($P$) and temperature ($T$) according to the equation:

$$\rho_{P,T} = \rho_0 [1 - \alpha(T - T_0)][1 + \beta(P - P_0)], \tag{8}$$

where $\rho_0$ is the standard density at $P_0 = 1 \times 10^5$ Pa and $T_0 = 298.15$ K, and $\alpha = 3.0 \times 10^{-5}$ K$^{-1}$ and $\beta = 1.0 \times 10^{-11}$ Pa$^{-1}$ are the coefficients of thermal expansion and compressibility, respectively.

The density model also takes into account phase transformations. Olivine transforms to wadsleyite and ringwoodite in the mantle transition zone[60] and to bridgmanite in the lower mantle[61]. Eclogitisation of the subducted basaltic and gabbroic crust is taken into account by linearly increasing the density of the crust with pressure from 0 to 16% in the $P$-$T$ region between experimentally determined garnet-in and plagioclase-out phase transitions in basalt[62].

### Resolution test

Model resolution mainly influence the initiation of the younger subduction (Supplementary Fig. 14). At half resolution (C817L1T5-HR), the younger subduction initiates and the model results remain as before. With double resolution (C817L1T5-DR), the younger subduction fails to initiate in some tests, and the possible reason is that resolution affects strain localisation due to resolution-dependence of strain weakening[63,64] (i.e., fine resolution with the same strain weakening parameters promotes strain localisation along the first/older fault structures). In order to verify this, in the subsequent test model (C817L1T5-DR-S), we exclude strain weakening and simplify the petrological model (by neglecting melting and hydration/dehydration), and as the result, the younger subduction initiates as in the reference model.

## Data availability

All kinematic observation and reconstruction data can be searched in the corresponding references. The modelling data used in this study are available in the OSF repository at https://doi.org/10.17605/osf.io/skcd2.

## Code availability

Geomaps are created by the Generic Mapping Tools (https://www.generic-mapping-tools.org/). The numerical code I2VIS is available upon request from Taras Gerya (taras. gerya@erdw.ethz.ch).

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

## Acknowledgements
Thanks to Thorsten W. Becker for providing plate motion and trench migration data and critical comments. J.L. acknowledge funding from Guangdong project 2017ZT07Z006 and NSFC projects (42222406, U1901214, 41974104), and T.G. acknowledge funding from SNSF Research Grant 200021_192296 and by ILP Task Force "Bio-geodynamics of the Lithosphere". Some of the colourmaps are from Scientific Colour Maps at http://www.fabiocrameri.ch/colourmaps. Numerical simulations were run with the clusters of the National Supercomputer Center in Guangzhou (Tianhe-II).

## Author contributions
K.Z. and J.L. conceived the initial idea and model; T.G. provided the numerical code and improved study design. K.Z. performed numerical experiments. All authors discussed the model results and wrote paper.

## Competing interests
The authors declare no competing interests.

## Additional information

**Peer review information** : *Nature Communications* thanks the anonymous, reviewers for their contribution to the peer review of this work. A peer review file is available.

