## [Peer Review File · Nature Communications]

REVIEWER COMMENTS

Reviewer #1 (Remarks to the Author):

I have read this manuscript with interest. The authors focus on a, so far, not overlooked tectonic process, observed in the geological record. Double subduction dynamics is not trivial and it makes sense to study this process using computer simulations. I am overall supportive of the publication after moderate revisions. Besides 4 moderate points, my comments mostly relate to the writing. Not everything was clear at the first read and I have several suggestions that I hope will help “waxing” the manuscript.

l. 14 - Waxing - I'm not sure waxing applies tot tectonic plates...

l. 29 - I'd call it 'setting' rather than 'region'. Regions other related to geographic locations (next sentence)

l. 35 - not clear what is meant by 'growth' of a plate. Also I would you 'delay' rather than 'retard'

l. 44-45 - the explanations are confusing. Why is obduction involved? is there overthrusting of an ocean on top of a continent? In the previous sentence the motion of overriding plate is controlled by surrounding subductions. In the sentence it's the motion of the overriding plate that seems to control subduction.

l. 60-61 - this reads weird, maybe a verb is missing?

Not sure what waxing of a plate means...Would be useful to clearly state what is meant by waxing and waning since it's also used on figure 1.

l. 82-85 (“We address, ...”) may be reformulated in a simpler way?

l. 96 “obduction” is not obvious

Moderate point: l. 188 the entire section "Plate motion reorganization in natural double subduction" is very interesting, I would actually place this section before the modelling. It provide essential observations, clarifies the wording and sets up the motivation for the modelling.

l. 256 "...However, the dynamics of plate and trench motion in the complex subduction systems are poorly understood" This reads like an introduction sentence not a conclusion one. Just remove it?

Moderate point: As a general comment to modelling study. A resolution study would be a great addition (run the reference with half resolution and with double resolution, then show the difference). This is essential for assessing the robustness and credibility of modelling results.

At that point, I now understand that "waxed" is active and "waned" is extinct.

l. 276 space between "5" and "km"

l. 288 do left/right plates have a finite length or are they infinite?

l. 298 Why an equation for surface processes and not for the remainder? The model sensitivity with regard to surface processes is not even investigated.

Moderate point: A lot of processes, that seemed unimportant while reading the main text, are here described. Do hydration/dehydration, melting, softening matter at all? Looks like the model is mostly controlled by imposed kinematics (plate velocities and weak zone activation) and buoyancy. I would be great to focus on the essential, not to confuse the readership.

Moderate point: Could a simple force balance analysis be applied? If yes, this would be a great addition. It would provide a key for the reader to understand the results. Looks like most important force are the horizontal push and the pull forces.

l. 357 "Thanks for the Thorsten W. Becker" —> Thanks to

l. 358 "Some of the colormaps from Scientific colour maps" misses a verb.

Reviewer #2 (Remarks to the Author):

In this study "Onset of Double Subduction Controls Plate Motion Reorganization", the authors develop a 2D numerical thermo-mechanical model for in-dip double subduction systems to show their evolution as a function of various physical parameters, such as convergence rates, overriding plate length and difference in initiation timing of the two subducting slabs. Their model results allows them to constrain the role of these variables in determining how a double-subduction system can sustain in time. The authors ultimately apply their simulation results to natural subduction zones: the Manila and Philippines double subduction system. The work is scientifically sound and impressive in terms of the new findings on the complex evolution of in-dip double subduction tectonics. The writing and figures are generally of high quality. Therefore, this manuscript is worthy of publication. However, there are some concerns (listed below), mainly pertaining to the initial boundary conditions of the numerical model and the applicability of the model to natural cases that need be addressed prior to consideration for its publication.

Specific points:

1. What are the evidences that the authors use as a basis of their arguments that the drastic decrease in the activity of the southern segment of the Manila subduction was solely a consequence of the initiation of the Philippines subduction? Please note that Philippines initiated after the Manila subduction with a gap of 15 Ma. Usually, most subduction zones show reducing trends in the convergence and trench retreat rates due to attainment of maturity after the initiation for different reasons, e.g. interaction with the 660 km boundary.
2. According to the authors, the difference between the initiation timings between the Manila and Philippines subduction is ~ 15 Ma. However, their own model results show that increase in the time difference between the onset of two subduction zones favors the formation of a single subduction system where the earlier subduction takes the lead role (Fig. 3e, Ln: 160-163). In their Fig. 3e, onset of the right subduction after $t = 7$ Ma, favors the formation of a single subduction although the right convergence velocity is 7 cm/year. Can the authors show a scenario in which double subduction would be dominated by the younger one for $t = 15$ Ma, as observed in the Manila and the Philippines settings? I wonder if in such a case, the convergence velocities will agree with what is observed in case of the Manila-Philippines double subduction. This is a major issue that surely needs to be resolved as it will validate the model applicability to natural scenarios.

3. In continuation of the earlier comment, can the slowdown of the Manila subduction as a consequence of the Philippines subduction initiation be quantifiably justified by providing geochemical/geophysical data?

4. Why do the authors choose to impose the velocities at a distance of 2000 km from the left and right model boundaries, instead of the actual model boundaries? They should explain the choice of this kind of kinematic condition in their modelling, as it is likely to influence the model results.

5. There are several numerical modelling work on in-dip double-subduction processes in recent time, which have also discussed the relative dynamics of the participating subduction zones. The authors should cite some of these studies, which can improve the completeness of this work.

Minor comments:

Ln 40: The word evolutionary does not suit this sentence. May be changed to “evolutionary”.

Ln 44-46: Does this happen in all the natural cases that the authors have discussed, or, is it a special scenario only applicable to certain cases? In any case, a reference would be ideal for this statement.

Ln 56-57: Same for the Central America and Lesser Antilles system. See the major comment.

Ln 91: Why do the authors claim that their subduction evolution is dynamic when they impose a kinematic velocity boundary condition in the model?

Ln 95: I think the “obduction” word is not suitable here. May considering changing to “overthrusting”?

Ln 111: Rephrase this sentence to better the clarity.

Ln 155: what does “it” refers to?

Ln 166-168: The sentence seems to be clunky. Please improve the phrasing.

Ln 324: add a space after 'cohesion'.

Ln 341: It should read as 'linearly increasing' or 'linear increase'.

Fig 4b: Change one set of data representation to dotted lines as both of them appear similar.

The authors are suggested to show a plot of the temporal change in the trench-retreat rates for both the subduction zones in Fig. 2. It can be added in the supplementary.

COMMENTS (C) & REPLIES(R) NCOMMS-23-11883

REVIEWER COMMENTS

Reviewer #1 (Remarks to the Author):

C: I have read this manuscript with interest. The authors focus on a, so far, not overlooked tectonic process, observed in the geological record. Double subduction dynamics is not trivial and it makes sense to study this process using computer simulations. I am overall supportive of the publication after moderate revisions. Besides 4 moderate points, my comments mostly relate to the writing. Not everything was clear at the first read and I have several suggestions that I hope will help “waxing” the manuscript.

R: We appreciate the constructive comments from the reviewer and made corrections accordingly and hope the “waxed” version is more readable.

C: l. 14 - Waxing - I’m not sure waxing applies to tectonic plates...

R: We double checked the terms “waxing and waning”. According to the Cambridge Dictionary, ‘wax and wane’ mean “to grow stronger and then weaker again”, which are often used to describe the grow and decline of the moon (from new moon to full moon, and vice versa). It also can be used in the phenomena of periodic changes, such as tide, glacier, climate and so on. Besides, ‘wax and wane’ also occur in the domains of Biology, Chemistry, Physics, Social Science, and, certainly, Earth Science. Geologists used ‘wax and wane’ to describe the intensity of plume and volcanism activity (Cande & Stegman, 2011) and dynamic topography (Chen et al., 2021). Here, we really want to use the terms to describe the ‘growing’ of younger subduction and the ‘declining’ of older subduction in the double subduction system and to emphasize the system established spatially and temporally in Earth that control plate motion reorganization. We explained the terms ‘wax and wane’ in the Abstract by adding ‘growing’ and ‘declining’ in brackets, respectively.

C: l. 29 - I’d call it ‘setting’ rather than ‘region’. Regions other related to geographic locations (next sentence)

R: We agree and use ‘setting’ in the manuscript.

C: l. 35 - not clear what is meant by ‘growth’ of a plate. Also I would you ‘delay’ rather than ‘retard’

R: Thanks for your advice. We have replaced ‘growth’ with ‘development’. The suggested term of ‘delay’ is indeed pretty good, and we find a much better term, i.e., ‘inhibit’, because we want to describe the influence of ‘delay’ and ‘suppress’.

C: l. 44-45 - the explanations are confusing. Why is obduction involved? is there overthrusting of an ocean on top of a continent? In the previous sentence the motion of overriding plate is controlled by surrounding subductions. In the sentence it’s the motion of the overriding plate that seems to control subduction.

R: Indeed, obduction is confusing. We have replaced all ‘obduction’ with ‘overthrusting’ to describe the contribution of the overriding plate on subduction. The partitioning of plate convergence will influence subduction patterns, i.e., passive subduction triggered by the overthrusting of the overriding plate versus active subduction driven by the motion of the subducting plate. Here, we want to emphasize that overthrusting of the overriding plate governs the evolution of the older subduction in the early stage. We modified the sentence (Lines. 51-55).

C: l. 60-61 - this reads weird, maybe a verb is missing?

R: With double check, we think the sentence is grammatically correct, and the verb ‘control’ is presented in line 61.

C: Not sure what waxing of a plate means...Would be useful to clearly state what is meant by waxing and waning since it’s also used on figure 1.

R: Agree. The terms of waxing and waning should be followed by subduction instead of plate. We checked the entire manuscript and made corrections.

C: l. 82-85 “We address, ...” may be reformulated in a simpler way?

R: We simplified the sentence to: “We address that, based on the modeling results, the onset and development of the younger subduction inhibit the older one through plate motion partitioning.” (Lines. 96-99)

C: l. 96 “obduction” is not obvious

R: We have corrected all ‘obduction’ with ‘overthrusting’.

C: Moderate point: l. 188 the entire section “Plate motion reorganization in natural double subduction” is very interesting, I would actually place this section before the modelling. It provide essential observations, clarifies the wording and sets up the motivation for the modelling.

R: Thanks for the comment, and we extracted a certain amount of text and imposed in the introduction. Regarding Figure 5, it is quite hard to place it before the modeling due to the following two reasons. Firstly, this figure contains not only observations but also modeling results (e.g., stress states marked by colored bars interpreted from model results). Secondly, we also want to predict the future evolution of these three examples, based on the past and present. Thus, we hope to keep this figure after modeling. We further improved the description of the motivation by referring text in this figure.

C: l. 256 “...However, the dynamics of plate and trench motion in the complex subduction systems are poorly understood” This reads like an introduction sentence not a conclusion one. Just remove it?

R: We modified this sentence as the following: “Complex subduction systems (e.g., double subduction zones) are likely established, governing regional geodynamical evolution.” (Lines. 317-318)

C: Moderate point: As a general comment to modelling study. A resolution study would be a great addition (run the reference with half resolution and with double resolution, then show the difference). This is essential for assessing the robustness and credibility of modelling results.

R.: We appreciate the suggestion and conducted additional runs testing model resolutions (see the results below). The new results suggest model resolutions mainly influence the initiation of the younger subduction. With half resolution (C817L1T5-HR), the younger subduction initiates and model results maintain the same as before. With double resolution (C817L1T5-DR), the younger subduction fails to initiate in some tests, and the possible reason is resolution affects strain localization due to resolution-dependence of strain weakening (Gerya, 2013; Lavier et al., 2000) (i.e., fine resolution with the same strain weakening parameters promotes strain localization along the first/older fault structures). In order to verify this, in the subsequent test model (C817L1T5-DR-S), we exclude strain weakening and simplify the petrological model (by neglecting melting and hydration/dehydration), and as the result the younger subduction initiates as in the reference model.

C: At that point, I now understand that “waxed” is active and “waned” is extinct.

R: (^-^) Cheers!

C: l. 276 space between “5” and “km”

R: Corrected.

C: l. 288 do left/right plates have a finite length or are they infinite?

R: The left and right plates have a finite length. Besides, we impose two weak zones at the left and right boundaries, to promote extension as a response to the divergent motion of the two plates. We added the description in the Method part (Lines. 375-377).

C: l. 298 Why an equation for surface processes and not for the remainder? The model sensitivity with regard to surface processes is not even investigated.

R: We added description of other conservation equations (mass, momentum, energy) to Methods We did not test previously the influence of surface processes since topography in our intra-oceanic subduction models is mostly submarine and is not subjected to erosion whereas sedimentation rate assumed to be very low. The approach of the surface processes we used in our model has been intensively tested and employed in previous studies (Gerya et al., 2021; Menant et al., 2020). In the revised paper, we tested model sensitivity with surface processes and show the results below. As expected, there is no significant difference in topography between the reference model (C817L1T5) and the new test model (C817L1T5-NS, without surface processes). Considering the role of surface processes is negligible, we delated the description about that.

C: Moderate point: A lot of processes, that seemed unimportant while reading the main text, are here described. Do hydration/dehydration, melting, softening matter at all? Looks like the model is mostly controlled by imposed kinematics (plate velocities and weak zone activation) and buoyancy. I would be great to focus on the essential, not to confuse the readership.

R: Agree. We simplified the Method (see **Method** part).

C: Moderate point: Could a simple force balance analysis be applied? If yes, this would be a great addition. It would provide a key for the reader to understand the results. Looks like most important force are the horizontal push and the pull forces.

R: We appreciate the suggestion and conducted force computation. We added the force balance analysis in the main text (Fig. 3). Force is computed by integrating normal stress vertically (see below) on the left and right plates. The force plots show the following features. In the early stage, plates in all models are under compression. After 5 Myr, the double subduction forms, and one can judge the dominant subduction by comparing F_{right} and F_{left} . If F_{right} is higher than F_{left} , the double subduction system is dominated by the younger one (Fig. 3i). Otherwise, the younger subduction is inhibited by the older one (Fig. 3j) or failed to initiate (Fig. 3k)

$$F_{\text{left/right}} = \int_{z_{\text{LAB}}}^{z_{\text{surf}}} \sigma_{xx} dz$$

C: l. 357 “Thanks for the Thorsten W. Becker” —> Thanks to

R: Corrected.

C: l. 358 “Some of the colormaps from Scientific colour maps” misses a verb.

R: Corrected.

REFERENCE

- Cande, S.C. & Stegman, D.R. 2011. Indian and African plate motions driven by the push force of the Reunion plume head. *Nature*, 475(7354):47-52. <http://dx.doi.org/10.1038/nature10174>.
- Chen, Y.W., Colli, L., Bird, D.E., Wu, J. & Zhu, H. 2021. Caribbean plate tilted and actively dragged eastwards by low-viscosity asthenospheric flow. *Nat Commun*, 12(1):1603. <http://dx.doi.org/10.1038/s41467-021-21723-1>.
- Gerya, T.V. 2013. Three-dimensional thermomechanical modeling of oceanic spreading initiation and evolution. *Physics of the Earth and Planetary Interiors*, 214:35-52. <http://dx.doi.org/10.1016/j.pepi.2012.10.007>.
- Gerya, T.V., Bercovici, D. & Becker, T.W. 2021. Dynamic slab segmentation due to brittle–ductile damage in the outer rise. *Nature*, 599(7884):245-250. <http://dx.doi.org/10.1038/s41586-021-03937-x>.
- Lavier, L.L., Buck, W.R. & Poliakov, A.N.B. 2000. Factors controlling normal fault offset in an ideal brittle layer. *Journal of Geophysical Research: Solid Earth*, 105(B10):23431-23442. <http://dx.doi.org/10.1029/2000jb900108>.
- Menant, A., Angiboust, S., Gerya, T., Lacassin, R., Simoes, M. & Grandin, R. 2020. Transient stripping of subducting slabs controls periodic forearc uplift. *Nat Commun*, 11(1):1823. <http://dx.doi.org/10.1038/s41467-020-15580-7>.

Reviewer #2 (Remarks to the Author):

C: In this study “Onset of Double Subduction Controls Plate Motion Reorganization”, the authors develop a 2D numerical thermo-mechanical model for in-dip double subduction systems to show their evolution as a function of various physical parameters, such as convergence rates, overriding plate length and difference in initiation timing of the two subducting slabs. Their model results allow them to constrain the role of these variables in determining how a double-subduction system can sustain in time. The authors ultimately apply their simulation results to natural subduction zones: the Manila and Philippines double subduction system. The work is scientifically sound and impressive in terms of the new findings on the complex evolution of in-dip double subduction tectonics. The writing and figures are generally of high quality. Therefore, this manuscript is worthy of publication. However, there are some concerns (listed below), mainly pertaining to the initial boundary conditions of the numerical model and the applicability of the model to natural cases that need be addressed prior to consideration for its publication.

R: We appreciate the constructive comments from the reviewer and made corrections accordingly. Regarding the reviewer’s main concerns (e.g., initial boundary conditions and model applicability to observations), we provide detailed explanations and thoroughly revised the manuscript.

Specific points:

C: 1. What are the evidences that the authors use as a basis of their arguments that the drastic decrease in the activity of the southern segment of the Manila subduction was solely a consequence of the initiation of the Philippines subduction? Please note that Philippines initiated after the Manila subduction with a gap of 15 Ma. Usually, most subduction zones show reducing trends in the convergence and trench retreat rates due to attainment of maturity after the initiation for different reasons, e.g. interaction with the 660 km boundary.

R: Good question. Firstly, we corrected the initiation of the Philippines subduction to 18 Ma. Thus, the gap between these two subductions is 10 Ma. In the submitted manuscript, we used 23 Ma as the initiation of the Philippines subduction (table 1 in (Gurnis et al., 2004); originally from (Yumul et al., 2003). The new age of the Philippines subduction initiation (i.e., 18 Ma) is more robust, mainly derived from SSZ ophiolite ((Huang et al., 2018) Figures 29, 30). Secondly, the Manila subduction is a passive subduction (i.e., driven by plate motion), and the subducted South China Sea plate was very warm and young (i.e., seafloor spreading started at 32 Ma (Yu et al., 2022)), unlikely to maintain self-sustaining subduction. Thirdly, the slab depths of the Manila subduction, the Negros subduction and the Cotabato subduction are 480 km, 390 km and 250 km, respectively (Wu et al., 2016), and there is no obvious interaction between the slabs and the 660 km boundary. Fourthly, we propose the “waxing and waning” of the two subduction zones mainly based on the present observations, for instance, the fastest trench retreat in the northern segment of the Manila subduction is nearly equal to the motion of the Philippines Sea Plate (Fig. 1), indicating plate motion/convergence dominates subduction in this region. Although we cannot exclude other possible reasons for the declining of the Manila subduction, we speculate the influence of the Philippines subduction (and our modeling results support this idea).

C: 2. According to the authors, the difference between the initiation timings between the Manila and Philippines subduction is ~15 Ma. However, their own model results show that increase in the time difference between the onset of two subduction zones favors the formation of a single subduction system where the earlier subduction takes the lead role (Fig. 3e, Ln: 160-163). In their Fig. 3e, onset of the right subduction after $t=7$ Ma, favors the formation of a single subduction although the right convergence velocity is 7 cm/year. Can the authors show a scenario in which double subduction would be dominated by the younger one for $t=15$ Ma, as observed in the

Manila and the Philippines settings? I wonder if in such a case, the convergence velocities will agree with what is observed in case of the Manila-Philippines double subduction. This is a major issue that surely needs to be resolved as it will validate the model applicability to natural scenarios.

R: Another good question. We give the following explanations. Firstly, as stated above, we revised the initiation time of the Manila subduction (from 23 Ma to 18 Ma). The gap becomes smaller, but it is still 10 Ma. Indeed, with 10 Ma interval, the younger subduction cannot initiate in the reference model (Fig. 2). However, we need to consider another important factor, i.e., plate convergence rate. With smaller convergence rates, the younger subduction can initiate with longer time intervals (see Fig. 3d for example, the top two cases: convergence rate is 4 cm/yr and the time interval is 10 Ma). Secondly, the convergence rate of the Manila subduction is 7-8 cm/yr in the present day, and we think the average convergence rate is much smaller (a simple calculation: slab length of ~500 km divided by 18 Ma gives 2.8 cm/yr). With smaller convergence rates, the younger subduction could initiate in a larger range of time intervals.

C: 3. In continuation of the earlier comment, can the slowdown of the Manila subduction as a consequence of the Philippines subduction initiation be quantifiably justified by providing geochemical/geophysical data?

R: Also see our replies to the above two comments. We propose that the development of the Philippines subduction inhibits the Manila subduction mainly based on the present observations. For instance, as shown by GPS data (Fig. 1), the convergence rate along the Manila trench decreases from 9 cm yr⁻¹ in the north (~21°N) to 5 cm yr⁻¹ in the south (~13°N, (Hsu et al., 2012). Meanwhile, the convergence rate along the northern Philippines trench is 4 cm yr⁻¹, indicating that the total convergence rate in the double subduction area (i.e., the southern Manila) equals to that in the single subduction area (i.e., the northern Manila). Thus, we propose that the Philippines subduction absorbs a certain amount of plate convergence, inhibiting convergence along the southern Manila trench. Regarding quantifying the slowdown process of the Manila subduction, it is really difficult to find the related geochemical/geophysical data. Plate reconstruction (e.g., (Sibuet et al., 2002) may provide some constrains, but the uncertainties are large.

C: 4. Why do the authors choose to impose the velocities at a distance of 2000 km from the left and right model boundaries, instead of the actual model boundaries? They should explain the choice of this kind of kinematic condition in their modelling, as it is likely to influence the model results.

R: Thanks for this comment and we added detailed explanations in the text. Basically, there are two common ways of imposing velocity boundary conditions on the side boundaries in geodynamic modeling. One is the standard way of imposing velocity conditions on the real side boundaries, and the other one is the internal velocity boundary condition as we employed. The one we used is also employed in previous studies (Perchuk et al., 2020) and it has the following advantages. (1) It gives more realistic mantle flow along the boundaries, since we do not need to impose opposite velocities (e.g., income and outcome of material along the side boundaries to carry out mass conservation) that force mantle flow. (2) Using the internal velocity boundary condition, models tend to run smoothly since no new material flows in through the side boundaries.

C: 5. There are several numerical modelling work on in-dip double-subduction processes in recent time, which have also discussed the relative dynamics of the participating subduction zones. The authors should cite some of these studies, which can improve the completeness of this work.

R: Thanks for this suggestion. We added a short review about double subduction modeling (Lines. 319-325).

Minor comments:

C: Ln 40: The word evolutionary does not suit this sentence. May be changed to “evolutionary”.

R: Corrected.

C: Ln 44-46: Does this happen in all the natural cases that the authors have discussed, or, is it a special scenario only applicable to certain cases? In any case, a reference would be ideal for this statement.

R: Yes, this phenomenon widespread occurred in face-to-face double subduction systems. We have rewritten this sentence and added the references (Lines. 45-50).

C: Ln 56-57: Same for the Central America and Lesser Antilles system. See the major comment.

R: Also see our replies to the above main comments. Basically, the three natural examples share the same logic. More detailed description is shown in the discussion part “Plate motion reorganization in natural double subduction”.

C: Ln 91: Why do the authors claim that their subduction evolution is dynamic when they impose a kinematic velocity boundary condition in the model?

R: Thanks for the comment and we give the following replies. Our thermomechanical coupled models are indeed geodynamical models, which means that the physical parameters of rocks (e.g., temperature, pressure, density, viscosity, stress) change dynamically. We solve Stokes equation for velocity and the kinematic velocity boundary condition is necessary for computing the linear equations. Regarding subduction evolution, in our opinion, it is dynamic rather than kinematic. The kinematic velocity imposed in our model represents plate motion as observed in nature, such as the Manila subduction driven by the motion of the Philippines Sea plate (i.e., without the push/overthrusting of the Philippines Sea plate, the South China Sea plate would not subduct since it was very young/buoyant). The main effect of the kinematic velocity is driving subduction initiation (providing horizontal compress). Afterwards, subduction (e.g., sinking rate, dipping angles, geometry) is dominated by the parameters of the subducting plate (e.g., plate age, rheology).

C: Ln 95: I think the “obduction” word is not suitable here. May considering changing to “overthrusting”?

R: Agree. We have replaced all “obduction” with “overthrusting”.

C: Ln 111: Rephrase this sentence to better the clarity.

R: We rephrased this sentence to: “....., resulting in the decrease in trench migration of the older subduction.” (Lines 125-130)

C: Ln 155: what does “it” refers to?

R: The term “it” refers to the younger subduction. We modified the sentence for clarity (Line 190-194).

C: Ln 166-168: The sentence seems to be clunky. Please improve the phrasing.

R: Agree. We rephrased the sentence to “Horizontal motion of the overriding plate initiates and drives the older subduction associated with trench migration in the early stage, prior to the onset of the younger subduction.” (Lines 205-209)

C: Ln 324: add a space after ‘cohesion’.

R: Corrected.

C: Ln 341: It should read as ‘linearly increasing’ or ‘linear increase’.

R: We corrected as ‘linearly increasing’.

C: Fig 4b: Change one set of data representation to dotted lines as both of them appear similar.

R: Corrected.

C: The authors are suggested to show a plot of the temporal change in the trench-retreat rates for both the subduction zones in Fig. 2. It can be added in the supplementary.

R: Here we plot trench-retreat rates over time, it seems as same as overridding plate mdotion of model C817L1T5 (model in Fig. 2). Actually, if there is no obvious deformation in the overriding plate (i.e., back-arc spreading), the trench-retreat rates could be equal to the overriding plate motion (Heuret & Lallemand, 2005). So the motion of overriding plate can indicate the trench retreat rate and we added the description in the main text.

REFERENCE

- Gurnis, M., Hall, C. & Lavier, L. 2004. Evolving force balance during incipient subduction. *Geochemistry, Geophysics, Geosystems*, 5(7). <http://dx.doi.org/10.1029/2003gc000681>.
- Heuret, A. & Lallemand, S. 2005. Plate motions, slab dynamics and back-arc deformation. *Physics of the Earth and Planetary Interiors*, 149(1-2):31-51. <http://dx.doi.org/10.1016/j.pepi.2004.08.022>.
- Hsu, Y.-J., Yu, S.-B., Song, T.-R.A. & Bacolcol, T. 2012. Plate coupling along the Manila subduction zone between Taiwan and northern Luzon. *Journal of Asian Earth Sciences*, 51:98-108. <http://dx.doi.org/10.1016/j.jseaes.2012.01.005>.
- Huang, C.-Y., Chen, W.-H., Wang, M.-H., Lin, C.-T., Yang, S., Li, X., Yu, M., Zhao, X., Yang, K.-M., Liu, C.-S., Hsieh, Y.-H. & Harris, R. 2018. Juxtaposed sequence stratigraphy, temporal-spatial

variations of sedimentation and development of modern-forming forearc Lichi Mélange in North Luzon Trough forearc basin onshore and offshore eastern Taiwan: An overview. *Earth-Science Reviews*, 182:102-140. <http://dx.doi.org/10.1016/j.earscirev.2018.01.015>.

Perchuk, A.L., Gerya, T.V., Zakharov, V.S. & Griffin, W.L. 2020. Building cratonic keels in Precambrian plate tectonics. *Nature*, 586(7829):395-401. <http://dx.doi.org/10.1038/s41586-020-2806-7>.

Sibuet, J.-C., Hsu, S.-K., Le Pichon, X., Le Formal, J.-P., Reed, D., Moore, G. & Liu, C.-S. 2002. East Asia plate tectonics since 15 Ma: constraints from the Taiwan region. *Tectonophysics*, 344(1-2):103-134. [http://dx.doi.org/10.1016/s0040-1951\(01\)00202-5](http://dx.doi.org/10.1016/s0040-1951(01)00202-5).

Wu, J., Suppe, J., Lu, R. & Kanda, R. 2016. Philippine Sea and East Asian plate tectonics since 52 Ma constrained by new subducted slab reconstruction methods. *Journal of Geophysical Research-Solid Earth*, 121(6):4670-4741. <http://dx.doi.org/10.1002/2016jb012923>.

Yu, M., Yumul, G.P., Dilek, Y., Yan, Y. & Huang, C.-Y. 2022. Diking of various slab melts beneath forearc spreading center and age constraints of the subducted slab. *Earth and Planetary Science Letters*, 579. <http://dx.doi.org/10.1016/j.epsl.2022.117367>.

Yumul, G.P., Dimalanta, C.B., Tamayo, R.A. & Maury, R.C. 2003. Collision, subduction and accretion events in the Philippines: A synthesis. *The Island Arc*, 12(2):77-91. <http://dx.doi.org/10.1046/j.1440-1738.2003.00382.x>.

REVIEWERS' COMMENTS

Reviewer #1 (Remarks to the Author):

The authors have taken great care to address all of my comments. I think manuscript has now substantially improved. There are still a few typos in the text which are listed below.

I. 460 - Thanks to the Thorsten W. Becker → Thanks to Thorsten W. Becker

I. 454 - Numerical datas → Numerical data

I. 224 - ... because of the vanish of overthrusting from the overriding plate. → ... because overthrusting from the overriding plate vanishes.

I. 180 - Forec → Force

Reviewer #2 (Remarks to the Author):

The revised version of the manuscript “Onset of Double Subduction Controls Plate Motion Reorganization” by Zhang et al. has significantly improved over the last version. The work has now become more concise and clear. This work can be a potential contribution to subduction studies. However, in my opinion, there are still some issues that need to be addressed for clarification and improvement of this study. Please find the main points of concern listed below. The modelling part of this manuscript seems to have improved and the initial queries have been answered to my satisfaction. However, doubts still remain over the application of the model results to natural cases, which should be addressed.

1. In the main text (Ln 58), the authors show that the initiation of Philippines subduction occurred at ~8 Ma, which is younger than the adjacent Manila subduction initiation at ~18 Ma. But, in the response the authors mention that they have reduced the initiation timing of the Philippines subduction by 5 Ma (23 Ma to 18 Ma) and have supported this argument with the SSZ ophiolite ages. This seems to be contradictory and needs clarification. Furthermore, the authors should provide more geological information to support the new initiation timing.

2. As the authors state that “Although we cannot exclude other possible reasons for the declining of the Manila subduction, we speculate the influence of the Philippines subduction”, it is important to discuss in the main text the other alternative processes that might have led to slowing of the manila subduction (e.g. Bina et al., 2020). They should at least discuss how the authors exclude these possibilities.

3. The authors have clarified that the convergence rate of the Manila subduction is 7 - 8 cm/yr in the present day, and they consider that the average convergence rate is much smaller than the present-day rate, as shown from a simple calculation: slab length of ~500 km divided by 18 Ma results in 2.8 cm/yr. Usually, subduction processes slow down with time leading to the waning of trench retreat rates and advancing velocities, which is evident from both geophysical observations and modelling studies. A simple example is the Indo-Eurasian system where convergence values decreased from ~15 cm/year to ~3 cm/year. However, in the case of Manila-Philippines subduction system, the convergence velocity seems to increase very significantly. For their "simple calculation" to work, the Manila subduction needs to be much slower during the initiation phase than that attained in the mature stage. This is another contradiction, which the authors need to clarify.

4. The authors are encouraged to go through the literature on in-dip double subduction modelling and cite them in the context of this study.

COMMENTS (C) & REPLIES(R) NCOMMS-23-11883A

REVIEWERS' COMMENTS

Reviewer #1 (Remarks to the Author):

C: The authors have taken great care to address all of my comments. I think manuscript has now substantially improved. There are still a few typos in the text which are listed below.

l. 460 - Thanks to the Thorsten W. Becker —> Thanks to Thorsten W. Becker

l. 454 - Numerical datas —> Numerical data

l. 224 - ... because of the vanish of overthrusting from the overriding plate. —> ... because overthrusting from the overriding plate vanishes.

l. 180 - Forec —> Force

R: Happy to hear that and thanks for the reviewer's careful checking. We corrected the mistakes addressed by the reviewer and double-checked the writing.

Reviewer #2 (Remarks to the Author):

C: The revised version of the manuscript “Onset of Double Subduction Controls Plate Motion Reorganization” by Zhang et al. has significantly improved over the last version. The work has now become more concise and clear. This work can be a potential contribution to subduction studies. However, in my opinion, there are still some issues that need to be addressed for clarification and improvement of this study. Please find the main points of concern listed below. The modelling part of this manuscript seems to have improved and the initial queries have been answered to my satisfaction. However, doubts still remain over the application of the model results to natural cases, which should be addressed.

R: Thank you very much for reviewing our revised manuscript. We reply to the reviewer’s new comments below.

C: 1. In the main text (Ln 58), the authors show that the initiation of Philippines subduction occurred at ~8 Ma, which is younger than the adjacent Manila subduction initiation at ~18 Ma. But, in the response the authors mention that they have reduced the initiation timing of the Philippines subduction by 5 Ma (23 Ma to 18 Ma) and have supported this argument with the SSZ ophiolite ages. This seems to be contradictory and needs clarification. Furthermore, the authors should provide more geological information to support the new initiation timing.

R: We mixed up the initiation timing of the Philippines subduction in the reply letter, and we are sorry about this. We corrected the initiation timing of the Manila subduction (from 23 Ma to 18 Ma), instead of the Philippines subduction (~8 Ma). The references supporting the new initiation timing of the Manila subduction are mentioned in the main text (Yu et al., 2022).

2. As the authors state that “Although we cannot exclude other possible reasons for the declining of the Manila subduction, we speculate the influence of the Philippines subduction”, it is important to discuss in the main text the other alternative processes that might have led to slowing of the Manila subduction (e.g. Bina et al., 2020). They should at least discuss how the authors exclude these possibilities.

R: Thanks for the suggestions, and we agree. We discussed briefly the possible influence of complex tectonic processes (e.g., collision, plume) and made changes in the main text (lines 274). Indeed, the continental collision occurred at the southern end of the Manila subduction, which may affect the subduction process (Bina et al., 2020). Previous 3D geodynamical models also suggest the possible lateral effect of the collision on subduction, i.e., forming curved subduction geometry (Magni et al., 2014; Moresi et al., 2014). However, we think that the collision located to the south of Manila subduction could only affect the southern tip of the Manila subduction based on the following two lines of evidence. Firstly, look at the southern Manila segment, only the southern tip is curved (anchored by the collision in the south), and the rest is straight (suggesting negligible effect from the collision). Secondly, the collision intensity in the southern end of the Manila subduction is very weak, evidenced by low surface relief and low convergence rate. To quantify the influence of the complex tectonic processes and the younger subduction in face-to-face double subduction systems scenario needs further study.

C: 3. The authors have clarified that the convergence rate of the Manila subduction is 7 - 8 cm/yr in the present day, and they consider that the average convergence rate is much smaller than the present-day rate, as shown from a simple calculation: slab length of ~500 km divided by 18 Ma results in 2.8 cm/yr. Usually, subduction processes slow down with time leading to the waning of trench retreat rates and advancing velocities, which is evident from both geophysical observations and modelling studies. A simple example is the Indo-Eurasian system where convergence values decreased from ~15 cm/year to ~3 cm/year. However, in the case of Manila-Philippines subduction system, the convergence velocity seems to increase very significantly. For their “simple calculation” to work, the Manila subduction needs to be much slower during the initiation phase than that attained in the mature stage. This is another contradiction, which the authors need to clarify.

R: This is a very important point, and we make the following clarification.

Firstly, the present convergence rate of the southern segment of the Manila subduction, which is overlapped with the Philippines subduction, is 3-4 cm/yr. The total convergence rate of the face-to-face double subduction is 7-8 cm/yr, which equals the convergence rate of the northern segment of the Manila subduction (i.e., the single subduction part). The slowing down (waning) of the Manila subduction is mainly evidenced by the comparison between the southern and northern Manila segments, i.e., the convergence rate (3-4 cm/yr) of the southern segment (double subduction) is much slower than that (7-8 cm/yr) of the northern segment (single subduction).

Secondly, the convergence rate of the Manila subduction is governed by (1) the Philippines Sea plate motion and (2) the onset of the Philippines subduction. It is quite difficult to compare the convergence rate of the Manila subduction at two different stages, since the Philippines Sea plate motion may change dynamically and temporally (i.e., the counter-clockwise rotation of the Philippines Sea plate leads to the acceleration of the subduction (Wu et al., 2016)). Instead, we should look at the two segments of the Manila subduction at the same stage, since the variation between the southern and northern segments is mainly by the onset of the Philippines subduction.

Thirdly, the convergence rate of the Manila subduction is governed by (1) the Philippines Sea plate motion and (2) the onset of the Philippines subduction. We should not compare the convergence rate of the Manila subduction at two different stages, since the Philippines Sea plate motion may change dynamically and

temporally (i.e., the counter-clockwise rotation of the Philippines Sea plate leads to the acceleration of subduction). Instead, we should look at the two segments of Malina subduction at the same stage, since the variation between the southern and northern segments is caused by the onset of the Philippines subduction (see point 2 above).

C: 4. The authors are encouraged to go through the literature on in-dip double subduction modelling and cite them in the context of this study.

R: We double-checked and added the following two related papers: Dasgupta and Mandal (2022); Lyu et al. (2019).

REFERENCE

- Bina, C.R., Cizkova, H. & Chen, P.F. 2020. Evolution of subduction dip angles and seismic stress patterns during arc-continent collision: Modeling Mindoro Island. *Earth and Planetary Science Letters*, 533. <http://dx.doi.org/ARTN.116054>
10.1016/j.epsl.2019.116054.
- Dasgupta, R. & Mandal, N. 2022. Role of double-subduction dynamics in the topographic evolution of the Sunda Plate. *Geophysical Journal International*, 230(1):696-713.
<http://dx.doi.org/10.1093/gji/ggac025>.
- Lyu, T., Zhu, Z. & Wu, B. 2019. Subducting slab morphology and mantle transition zone upwelling in double-slab subduction models with inward-dipping directions. *Geophysical Journal International*, 218(3):2089-2105. <http://dx.doi.org/10.1093/gji/ggz268>.
- Magni, V., Faccenna, C., Van Hunen, J. & Funiciello, F. 2014. How collision triggers backarc extension: Insight into Mediterranean style of extension from 3-D numerical models. *Geology*, 42(6):511-514. <http://dx.doi.org/10.1130/g35446.1>.
- Moresi, L., Betts, P.G., Miller, M.S. & Cayley, R.A. 2014. Dynamics of continental accretion. *Nature*, 508(7495):245-248. <http://dx.doi.org/10.1038/nature13033>.
- Wu, J., Suppe, J., Lu, R. & Kanda, R. 2016. Philippine Sea and East Asian plate tectonics since 52 Ma constrained by new subducted slab reconstruction methods. *Journal of Geophysical Research-Solid Earth*, 121(6):4670-4741. <http://dx.doi.org/10.1002/2016jb012923>.
- Yu, M., Yumul, G.P., Dilek, Y., Yan, Y. & Huang, C.-Y. 2022. Diking of various slab melts beneath forearc spreading center and age constraints of the subducted slab. *Earth and Planetary Science Letters*, 579. <http://dx.doi.org/10.1016/j.epsl.2022.117367>.